# Intradevice Repeatability and Interdevice Comparison of Two Specular Microscopy Devices in a Real-Life Setting: Tomey EM-4000 and Nidek CEM-530

**DOI:** 10.3390/medicina60071110

**Published:** 2024-07-09

**Authors:** Mateusz Kecik, Martina Kropp, Gabriele Thumann, Bojan Pajic, Josef Guber, Ivo Guber

**Affiliations:** 1Department of Ophthalmology, University of Geneva, 1205 Geneva, Switzerland; mateusz.kecik@hcuge.ch (M.K.); martina.kropp@unige.ch (M.K.); gabriele.thumann@hcuge.ch (G.T.); bojan.pajic@orasis.ch (B.P.); 2Experimental Ophthalmology, University of Geneva, 1205 Geneva, Switzerland; 3Eye Clinic ORASIS, Swiss Eye Research Foundation, 5734 Reinach, Switzerland; 4Faculty of Sciences, Department of Physics, University of Novi Sad, Trg Dositeja Obradovica 4, 21000 Novi Sad, Serbia; 5Faculty of Medicine of the Military Medical Academy, University of Defense, 11000 Belgrade, Serbia; 6Department of Ophthalmology, University of Basel, 4001 Basel, Switzerland; josef.guber@kssg.ch

**Keywords:** corneal endothelium, specular microscopy, anterior segment, Fuchs’ dystrophy

## Abstract

*Background and Objectives*: The purpose of this study was to compare two commercially available specular microscopes (Tomey EM-4000 and Nidek CEM-530) in a real-life clinical setting in terms of intra- and interdevice variability. The study was conducted on all patients seen in a clinical practice specializing in anterior segment pathologies, regardless of the purpose of their visit. *Materials and Methods*: In total, 112 eyes of 56 patients (age 23–85 years old) were included in the study. Each eye was measured three times with each device (for a total of six measurements), and results for central corneal thickness (CCT) and corneal endothelial cell density (ECD) were recorded. The results were then evaluated with the D’Agostino–Pearson normality test and compared with a Wilcoxon signed-rank test, *t*-test, ANOVA or Mann–Whitney test for intra- and interdevice variability. *Results*: Both specular microscopes produced very reliable reproducible intradevice results: The Tomey EM-4000 measured an ECD of 2390 ± 49.57 cells/mm^2^ (mean ± standard error of mean); the range was 799–3010 cells/mm^2^. The determined CCT was 546 ± 5.104 µm (mean ± standard error of mean [SEM]); the range was 425–615 µm. The measurements with the Nidek CEM-530 revealed an ECD of 2417 ± 0.09 cells/mm^2^ (mean ± SEM); the range was 505–3461 cells/mm^2^ (mean ± SEM). The mean CCT detected was 546.3 ± 4.937 µm (mean ± SEM); the range was 431–621 µm. The interdevice differences were statistically significant for both parameters, ECD (*p* = 0.0175) and CCT (*p* = 0.0125) (*p* < 0.05). *Conclusions*: The Nidek CEM-530 and the Tomey EM-4000 both produced reliable and reproducible results in terms of ECD and CCT. The absolute measurements were statistically significantly different for CCT and ECD for both devices; the Nidek produces slightly higher values.

## 1. Introduction

The human corneal endothelium is composed of a monolayer of polygonal (mostly hexagonal)-shaped cells located beneath Descemet’s membrane [1]. In its physiological state, each cell measures about 5–6 μm in thickness and 20 μm in diameter [2]. The corneal endothelium is arranged in a regular, honeycomb-like pattern, with cells tightly packed together [2]. Intercellular junctions include tight junctions, which seal the gap between adjacent cells and regulate the transport of water, ions and molecules through the paracellular pathway that helps to maintain the integrity of the endothelial barrier while preventing the fluid from leaking into the stroma [1,2]. These tight junctions provide a weak resistance to paracellular movement of water and solutes, thus allowing a passive diffusion of fluid and nutrients from the anterior chamber, through the endothelial cell and into the overlaying corneal stroma [1]. Conversely, endothelial cells employ active transport mainly via sodium/potassium ATPase transporters in order to move fluid against the osmotic gradient and back into the anterior chamber [1]. These two mechanisms create a balance between passive diffusion and active removal of fluid within the corneal endothelium and have been called the pump–leak model [1]. Maintenance of this balance is the main function of corneal endothelial cells, and thus, these cells regulate the movement of fluid and nutrients between the cornea and the aqueous humor [1]. By actively pumping fluid out of the stroma, endothelial cells create a pressure gradient that draws fluid out of the cornea and prevents it from swelling, maintaining corneal transparency [1]. The endothelium also plays a role in the metabolism and transport of nutrients like glucose, lactate and ascorbic acid from the aqueous humor to the cornea, thus regulating the metabolic activity of the corneal epithelium and stroma, which are dependent, either partially like the former or totally like the latter, on the corneal endothelium for their oxygen and nutrient supply [2].

The monolayer of hexagonal corneal endothelial cells thus plays a chief role in maintaining corneal transparency by regulating corneal hydration and nutrition; without metabolic pump and an inner barrier, the corneal stroma would become quickly hydrated and would lose its transparency. The ideal hydration level to achieve optimal spacing of collagen fibrils and maintain corneal clarity is 78% [3]. Humans are born with around 5000 endothelial cells/mm^2^ [4]. This number drops to 3000 cells/mm^2^ with corneal growth and apoptosis in the late teens [5]. Thereafter, the number of corneal endothelial cells undergoes a steady decrease of 0.3–0.6%/year [5]. Ethnic differences exist in central endothelial cell density, with African American, Japanese and Chinese populations having a larger cell count and Indian populations a lower cell count compared to Caucasians [6,7]. Factors like trauma, cataract surgery or genetic predispositions, like Fuchs’ endothelial dystrophy, lead to a non-physiologic loss of endothelial cells and possibly loss of corneal transparency [2,3,8]. Traditionally, it has been believed that endothelial cells do not regenerate, but there exists an adult stem cell population near the Schwalbe’s line that demonstrates a limited cell proliferation property after injury [9]. Although no cut-off value exists for corneal decompensation, it usually occurs at 750–500 cells/mm^2^ under physiological intraocular pressure (IOP) [10]. In addition to density, endothelial cell hexagonality is another important parameter of endothelial cell health; with cell loss, the remaining cells increase in diameter, elongate and spread in order to cover empty space and maintain a uniform cell monolayer, but they lose their hexagonality in the process [5]. Theoretically, an ideal human cornea would comprise 100% hexagonal cells, but in practice, this number is around 60% in healthy adults [11]; this number decreases in diseased corneas [12].

Clinically, several methods to assess corneal endothelial cell health exist. Upon taking patient history, vigilance must be exercised towards any symptoms related to morning blur, which can signify endothelial cell compromise. Slit lamp biomicroscopy can reveal diffuse or local areas of corneal thickening and guttae, which are localized areas of Descemet’s membrane excrescences and cell loss [12]. Confocal microscopy can visualize individual endothelial cells, their quantity and morphology. Those methods rely, however, on patients’ subjective symptoms or ophthalmologist subjective assessment, or they require expensive and difficult-to-master confocal microscopes [13]. Clinicians, on the other hand, require dependable and precise devices in order to be able to quantify and monitor the number and features of corneal endothelial cells in vivo without subjective bias; the method most broadly used in clinical practice today is specular microscopy. It is a noninvasive diagnostic modality for imaging the corneal endothelium, based on the original design of a laboratory microscope by Maurice [14], using an optical reflection microscope focused on endothelial cells and specularly reflected rays focused on a detector. The specular reflex occurs at a smooth and regular interface between two structures of different refractive indices where the light from the subject has an angle of incidence equal to the angle of reflection to the observer. Specular microscopy of endothelial cells is possible because the refractive index of the endothelial cells is greater than that of the aqueous humor (1.336), enabling the endothelial cells to reflect 0.022% of the projected light [15]. The first reports of using the specular reflex from a slit lamp to see the corneal endothelium come from the 1920s [11]. The area of the specular reflex is dependent on the radius of curvature of the reflecting surface; light reflected from a flat surface will reproduce the area of the light source, while light reflected from a cylinder will be condensed 90° to its axis [11]. Another physical restriction to the size of the imaged area is caused by the proximity of the two concentric surfaces or, in the case of the cornea, the epithelium and the endothelium [11]. Due to the fact that the corneal epithelium has a high refractive index difference between air and the tear/epithelium, the beam of light that passes through the cornea is reflected off both the epithelial and endothelial interface [11]. This fact means that the viewable specular area is a rectangle with the radius of corneal curvature dominating its height [11]. The slit beam of light passing through the corneal stroma is also affected by scatter from the collagen lamellae and keratocytes, which reduces the contrast of the endothelial cell image; this effect is worsened by increasing the width of the beam of light and explains why the left side of the image has better contrast as it is less affected by stromal scatter [11].

Specular microscope designs can be divided into corneal epithelial cell layer contact or non-contact instruments. The contact instruments have an objective microscope lens that applanates the corneal surface and require a topical anesthetic, while the non-contact designs rely on automatic image focusing technology [11]. The applanating specular microscopes flatten the corneal curvature and thus enlarge the specular reflex area, but both designs are limited by the bright epithelial reflection surface and the corneal thickness [11]. Consequently, the width of the viewable rectangle of endothelial cells with both specular microscopes is equivalent.

Modern specular microscopes calculate corneal thickness as well as endothelial cell density/mm^2^ and their pleomorphism (percentage of hexagonal cells) and polymegatism (variability of cell area). The devices can monitor both central and peripheral areas, but in clinical practice, most measurements are taken in the pupillary area of the cornea [16]. Herein lies a difficulty as a specular photograph of a single 0.25 mm × 0.54 mm area of the cornea may not be representative of the whole structure [16]. Ideally, sampling of over 20% of the endothelial surface is necessary to provide an accurate characterization of cell health [16].

Being able to efficiently and reliably quantify and monitor corneal endothelial cells is an important aspect in different clinical settings such as of approval of ophthalmic drugs and surgical devices, especially minimally invasive glaucoma surgeries [17]. For instance, the CyPass supraciliary microshunt (Alcon Laboratories, Inc) device had to be withdrawn from the market because of the accelerated loss of cells in operated patients [18]. Although there were no cases of corneal decompensation requiring device explantation and an endothelial graft, the progressive loss of cells could first be detected thanks to specular microscopy, and a voluntary withdrawal from the market was made. Other clinical implications are preoperative cataract assessment, follow-up of endothelial diseases such as Fuchs’ endothelial dystrophy and postoperative evaluations after posterior lamellar keratoplasty, i.e., DMEK [12,19].

Fuchs’ endothelial corneal dystrophy is a progressive disease that affects the corneal endothelium, leading to cell loss and in turn affecting corneal hydration and transparency [12]. It is the most common cause of endothelial cell dysfunction and corneal decompensation with the prevalence of 3–11% and is the leading indication of keratoplasty in the United States [8]. The pathogenesis of Fuchs’ endothelial corneal dystrophy is not yet fully understood, but it is an autosomal-dominant and oxidative stress disorder [12]. Several causal genetic mutations and single-nucleotide polymorphisms have been identified [12,20]; they make the corneal endothelium more vulnerable to oxidative stress and environmental factors. Current treatment options include corneal transplantation and endothelial keratoplasty [12]. Fuchs’ endothelial corneal dystrophy typically affects individuals over the age of 40–50 years old and is more common in women than men [12]. The disease is usually bilateral but may be asymmetric [12]. In the early stages of the disease, patients may not experience any symptoms, but as it progresses, usual complaints include blurred vision, glare and halos around lights, especially early in the morning right after waking up [12]. Later, in advanced stages of the disease, the cornea may develop bullae, leading to pain, discomfort and eventually to subepithelial scarring [12]. Even in the early stages of the disease, the corneal guttae may cause intraocular light scatter and induce high-order optical aberrations that can be very debilitating for the patient despite the absence of corneal edema [21]. The diagnosis of Fuchs’ endothelial corneal dystrophy is based on ophthalmic clinical examination and corneal imaging. Slit lamp biomicroscopy can reveal the presence of guttae, corneal edema and bullae [12]. Corneal imaging, including anterior segment optical coherence tomography, can provide detailed information about the corneal endothelium and Descemet’s membrane [12]. Specular microscopy is usually the exam of choice in monitoring cell counts in patients with Fuchs’ endothelial corneal dystrophy [19].

Secondary corneal endothelial dysfunction mainly occurs after an intraocular surgical procedure, retained lens fragment, medication toxicity or a penetrating corneal injury [3]. Many systemic drugs have been described to cause corneal endothelial cell dysfunction and corneal edema, including amantadine [22], methylphenidate [23], digoxin [24] and erlotinib [25]. Topical medications associated with corneal endothelial damage are mostly carbonic anhydrase inhibitors, although these are only clinically significant in eyes with a preexisting fragile endothelium [17]. The previously mentioned CyPass supraciliary microshunt withdrawal is a good example of the necessity of a long follow-up with specular microscopy after intraocular device implantation. Although similar rates of endothelial cell loss were measured between the microshunt phacoemulsification and phacoemulsification-only groups at 2 years (12.0% and 8.7%, respectively), an extension safety study observed endothelial cell losses of 20.4% and 10.1% at 5 years, respectively [18]. Modern phacoemulsification techniques have greatly reduced the risk of pseudophakic bullous keratopathy, which was more prevalent especially with iris- or angle-supported intraocular lenses [26]. Today, most cases of pseudophakic bullous keratopathy after uncomplicated cataract surgery with modern intraocular lens are usually related to corneal endothelial fragility and low preoperative endothelial cell counts [27], emphasizing the importance of preoperative screening and specular microscopy in the event of a suspected endothelial weakness.

In this study, we compared two commercially available non-contact specular microscopes, the Tomey EM-4000 (Tomey, Nürnberg, Germany) and the Nidek CEM-530 (Nidek, Tokyo, Japan), for intradevice repeatability and interdevice comparison when measuring corneal thickness and endothelial cell density in a real-life setting.

## 2. Materials and Methods

The study was conducted in a clinical practice in a real-life setting on patients presenting for ophthalmic examination before planned cataract surgery. The inclusion criteria were age over 18 years old and the provision of informed consent for participation in the study. The exclusion criteria were an age of less than 18 years old, inability to provide informed consent and a significant anterior segment pathology preventing visualization of endothelial cells through specular microscopy.

All patients underwent full clinical examination including the measurement of best-corrected visual acuity (BCVA), slit lamp, IOP measurement by either air-puff or Goldmann applanation tonometry and dilated fundoscopic examination performed by a trained ophthalmologist (I.G.), and 3 repeated measurements with each specular microscope were taken by a trained technician.

In total, 112 eyes of 56 patients were included in the study; the mean age of the participants was 61.05 ± 18.53 years old (range 22–85), 24 participants were female (42.9%) and 32 were male (57.1%). Four patients were examined with the Nidek only. A total of 668 measurements were taken; 25 measurements taken with the Tomey and 3 taken with the Nidek were excluded for poor quality, leaving 640 measurements for analysis. The results can be seen in Table 1.

The collected results were statistically analyzed by masked researchers unrelated to the clinic using the GraphPad Prism software (version 9.5.1 (733)). The statistical tests employed were a Friedman test, ANOVA, a Wilcoxon signed-rank test, a Mann–Whitney test and a paired *t*-test according to the sample distribution determined by performing a D’Agostino Pearson normality test. The results are shown as mean ± SEM if not mentioned otherwise.

The study adhered to the principles of the Declaration of Helsinki and were approved by the cantonal ethical commission of Zurich, Switzerland, number 2022-01970.

### 2.1. Tomey EM-4000 (Tomey, Nürnberg, Germany)

Measurement with this device, called “Tomey” in the following, is performed in a non-contact fashion (it usually takes 4 s for each eye) with central auto-alignment, including automatic analysis of the endothelium. The Tomey has an integrated non-contact pachymetry (Figure 1).

### 2.2. Nidek CEM-530 (Nidek, Tokyo, Japan)

Similar to the Tomey, the measurements with this device, called “Nidek” in the following, are performed in a non-contact manner, the difference being that the alignment is performed manually and the measurement thereafter is self-triggered, including the automatic analysis of the endothelium. The Nidek also has an integrated non-contact pachymetry (Figure 2).

## 3. Results

Both specular microscopes produced consistent intradevice results (Figure 3): Tomey EM-4000 measured an ECD of 2390 ± 49.57 cells/mm^2^ (range 799–3010 cells/mm^2^) and a CCT of 546 ± 5.104 µm (range 425–615 µm), while the Nidek CEM-530 measured an ECD of 2417 ± 0.09 cells/mm^2^ (range 505–3461 cells/mm^2^) and a mean CCT of 546.3 ± 4.937 µm (range 431–621 µm). The results and patient demographics can be seen in Table 1.

The mean of the consecutive measurements in terms of ECD for the Tomey were 2406 ± 355, 2380 ± 375 and 2380 ± 369 cells/mm^2^ and 2389 ± 261, 2377 ± 270 and 2392 ± 286 cells/mm^2^ for the right and left eye, respectively, and for the Nidek, they were 2480 ± 481, 2477 ± 487 and 2439 ± 509 cells/mm^2^ and 2522 ± 378, 2508 ± 412 and 2514 ± 416 cells/mm^2^, respectively. The mean consecutive CCT measurements for the Tomey were 531.2 ± 35.72, 531.2 ± 35.98 and 531.4 ± 35.79 µm and 522.1 ± 33.89, 521 ± 34.22 and 528 ± 34.40 µm for the right and left eye, respectively, and for the Nidek, they were 545.8 ± 37.36, 545.7 ± 37.40 and 545.8 ± 37.36 µm and 545.5 ± 35.38, 545.7 ± 35.90 and 545.7 ± 35.72 µm, respectively. Overall, the intradevice measurements were in statistically significant accordance for each device and eye (Tomey OD vs. OG *p* = 0.0796; Nidek OD vs. OG *p* = 0.9910; Tomek measurement 1–3 *p* = 0.7972; Nidek measurements 1–3 *p* = 0.6207).

On the other hand, the interdevice variability in terms of differences in measurements between devices (Figure 4) was statistically significant for both the CD counts (*p* = 0.0175) and for CCT (*p* = 0.0125). This represented a difference of 4.87% and 3.07% in terms of ECD and CCT, respectively.

## 4. Discussion

This study validates ECD and CCT measurements with the Tomey EM-4000 and Nidek CEM-530 devices in a real-life scenario determining intra- and interdevice variability. Both specular microscopes produced reliable results with little intradevice variation and clinically acceptable interdevice conformity.

Caution must be exercised, as the Nidek CEM-530 had a tendency to yield higher ECD values. Even though the difference was small, it was within the statistically significant range and should be taken into account when following patients with different devices or when switching to another device. It might be recommended to use the same device when performing studies in order to obtain reliable follow-up results. Both specular microscopes have similar endothelial capture fields (the Tomey has a photographic range of 0.25 mm × 0.54 mm and the Nidek’s is 0.25 mm × 0.55 mm). The Nidek’s slightly larger field could account for the discrepancies in measurements between devices; as the average diameter of a human corneal endothelial cell is 20 μm or 0.02 mm, it is able to include more cells in the photograph for analysis. Another well-known factor accounting for differences in endothelial cell density in measurements with specular microscopy is the relatively small size of the capture field in relation to the total endothelial cell area; it accounts for some ECD variation, as even with automatic centering, no specular photograph is microscopically aligned with the previous one when taking repeated measurements [16,19]. Moreover, due to local variability in cell density, in order to assess the complete clinical picture, at least 20% of the endothelial cell surface should be imaged; this is currently out of the scope of commercially available specular microscopes [16], leading to some authors suggesting to perform repeated and peripheral corneal as well as central cell counts [16]. Another reason for this difference might be related to how the devices recognize and calculate cells at the edge of the studied frame (Figure 5). Endothelial cells that are not entirely included in the testing area might be chosen by the automated algorithm to be either included or excluded from the total number of measured cells, potentially leading to different results. The difference between both devices is relatively small (4.87%), and an argument can be made that this bears little clinical significance. Indeed, a study on corneal donor tissues proposed 5% as a threshold of clinical significance for endothelial cell counts.

The CCT values were also technically statistically significantly different, but here, the difference was even lower at 3%, which has very little significance in the clinic.

In our study, all measurements were performed by a trained medical nurse. This is a very important factor in maintaining an efficient daily routine, as ophthalmologists with full consultation lists and limited time can reliably delegate the endothelial cell count measures to paramedical staff. Measurements were performed with central fixation in an automated fashion; both devices use a joystick to center onto the patient’s eye. While the Tomey is completely self-triggered with auto-alignment, the Nidek needs to be manually aligned before the subsequent measurement is auto-triggered as it is with the Tomey. We found that with the Tomey, a total of 25 measurements had to be excluded compared to 3 with the Nidek, which is, in our opinion, due to the auto-alignment function of the Tomey. The auto-alignment function tended to fail in eyes with unstable fixation, uncooperative patients or eyes with discrete nystagmus. Fortunately, in such cases, there is a possibility to perform the measurement manually, but this usually requires the presence of a physician. In our study, we chose to exclude those measurements from the analysis, as we intended to study results obtained in a fashion that is most frequently used in clinical practice and reflects real-world practices best.

To our knowledge, this is the first study in the literature to compare these two devices in terms of inter- and intradevice repeatability. There exist studies on different models and brands of specular microscopes demonstrating statistically significant differences in measurements, leading to authors recommending the user to follow patients with one device only [28,29,30,31,32,33,34]. The differences in ECD and CCT between the Tomey and Nidek were statistically significant in our study, and therefore, we recommend caution when switching between specular microscopes in a clinical setting.

The strengths of the study include its real-life setting including patients typically encountered in clinical practice (22–85 year old) comprising both cases with physiologic corneas (cell counts over 3400 cells/mm^2^) as well as those with low endothelial cell counts (as low as 500 cells/mm^2^). Further strengths include the fact that six measurements were performed per patient entirely by paramedical staff, as is the norm in many centers, and the results were analyzed by masked researchers unrelated to the clinic and masked to the devices used in the study.

A relative weakness is the absence of patients with bullous keratopathies and highly decompensated corneas, as the highest measured corneal thickness in the study was 621 µm. They, however, represent outliers and are a well-known technological limit for specular microscopy, which is unreliable and usually even impossible in those patients; for those reasons, we chose not to include such corneas in the study. The study also only examined two parameters provided by specular microscopes (ECD and CCT), which are most widely used by clinicians and in scientific reports; further studies could include other parameters such as the polymegathism (coefficient of variation), pleomorphism and cell area.

## 5. Conclusions

In conclusion, both specular microscopes provide high-quality and reliable results in terms of ECD and CCT in a real-life setting. There was a statistically significant difference between the two devices both in terms of the ECD count and CCT, with the Nidek providing higher values. Therefore, some care must be taken when comparing results from one to another device, especially in multicenter studies or in clinical centers where both devices are available.

## Figures and Tables

**Figure 1 medicina-60-01110-f001:**
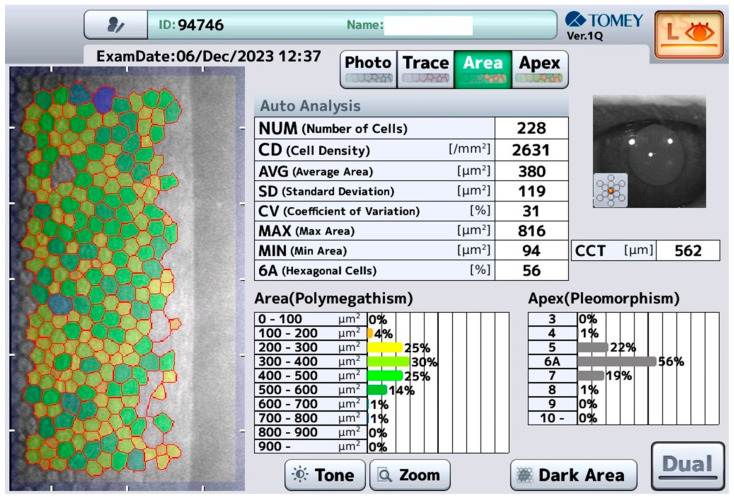
Typical view of the endothelial cell count performed with the Tomey device. NUM—number of cells evaluated in the scan, CD—cell density, AVG—average cell area, SD—standard deviation of the cell area, CV—coefficient of variation, MAX—area of the largest cell, MIN—area of the smallest cell, 6A—number of cells with a hexagonal shape.

**Figure 2 medicina-60-01110-f002:**
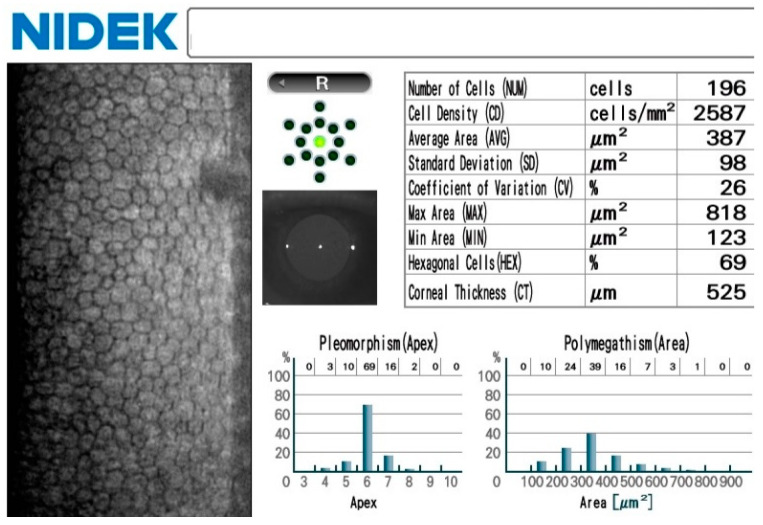
Typical results of an endothelial cell count performed with the Nidek device.

**Figure 3 medicina-60-01110-f003:**
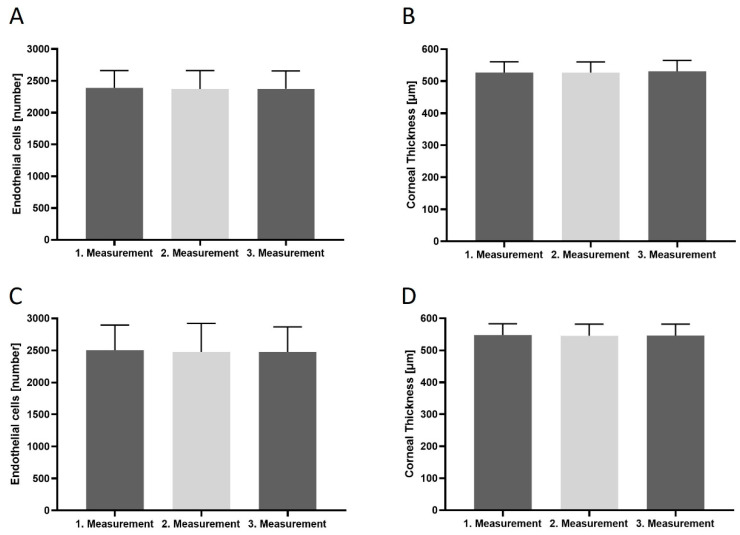
(**A**–**D**) Intradevice variability in terms of endothelial cell counts and central corneal thickness, showing close agreement in both measured parameters for both devices. The graphs represent each of the 3 consecutive measurements with each device.

**Figure 4 medicina-60-01110-f004:**
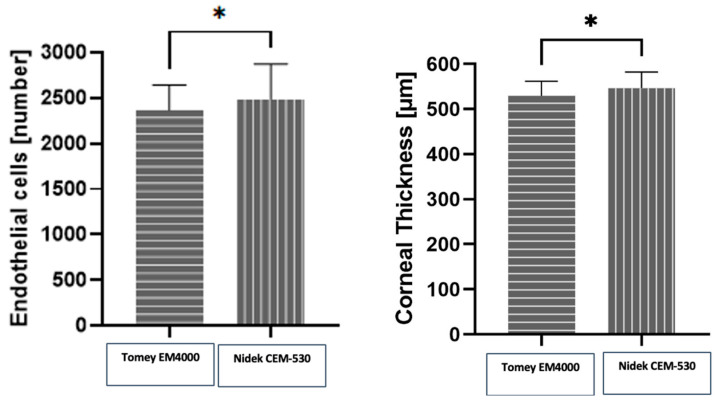
Interdevice variability in terms of endothelial cell count and central corneal thickness showed a reduced endothelial cell count with the Tomey compared to the Nidek device with an average of 121.2 cells (4.87%) less (* *p* = 0.0175). Also, measured central corneal thickness was lower in the Tomey with a difference of 16.75 µm (3.07%) (* *p* = 0.0125).

**Figure 5 medicina-60-01110-f005:**
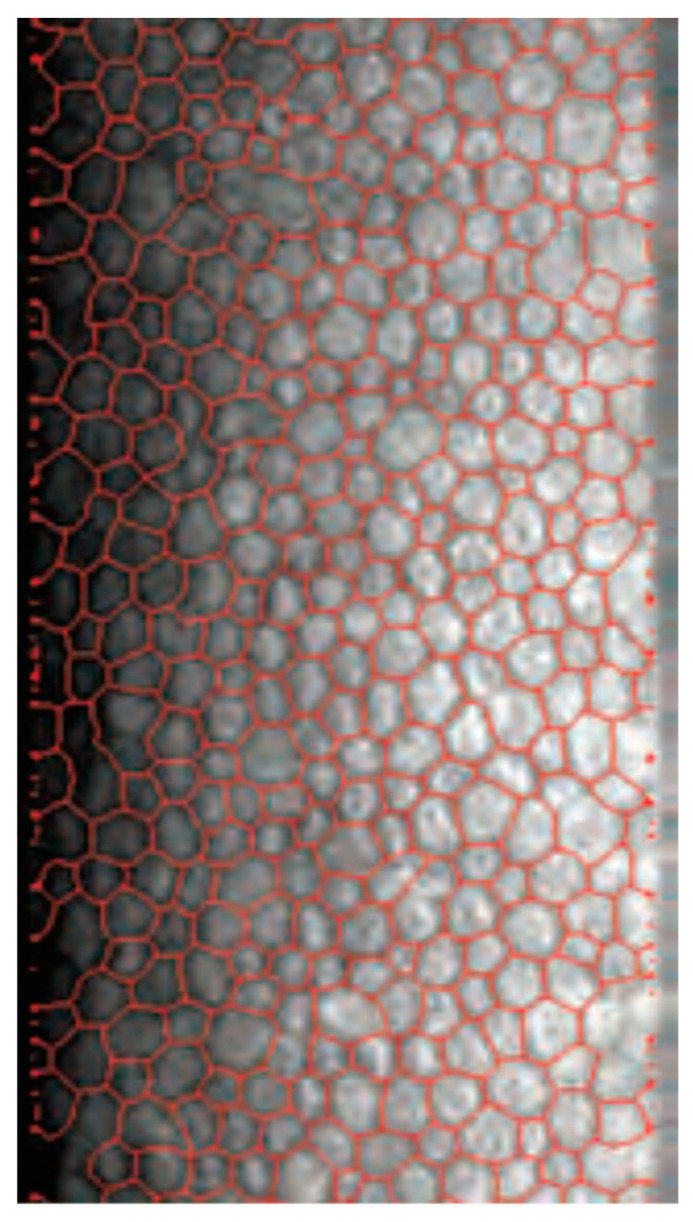
Example of specular microscopic image with endothelial cell segmentation performed with Tomey.

**Table 1 medicina-60-01110-t001:** Baseline demographics, clinical characteristics and endothelial cell and central corneal thickness measurements of the study population. CD—endothelial cell density, CCT—central corneal thickness.

Demographic and Clinical Data	Mean ± SD (%)
Age (years; mean ± SD)	61.05 ± 18.53
Range	22–85
Gender	
Female	24 (42.9%)
Male	32 (57.1%)
Number of eyes included	112
Number of measurements	668
Excluded for poor quality	28
Tomey EM-4000	25
Nidek CEM-530	3
Measurements included in analysis	640
Results	
Tomey EM-4000	
CD (cells/mm^2^)	2390 ± 49.57
Range	799–3010
CCT (μm)	546 ± 5.104
Range	425–615
Nidek CEM-530	
CD (cells/mm^2^)	2417 ± 80.09
Range	505–3461
CCT (μm)	546.3 ± 4.94
Range	431–621

## Data Availability

The raw data supporting the conclusions of this article will be made available by the authors on request.

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
