# Peer review of "Intradevice Repeatability and Interdevice Comparison of Two Specular Microscopy Devices in a Real-Life Setting: Tomey EM-4000 and Nidek CEM-530"

_medicina, 2024, doi:10.3390/medicina60071110_

Round 1
Reviewer 1 Report
Comments and Suggestions for Authors
The purpose of this study was to compare two commercially available specular microscopes (Tomey EM-4000 and Nidek CEM-530) in a real-life clinical setting in terms of intra- and inter-device variability. The paper is very clearly written, the conclusions are consistent with the provided evidence. I have nothing to add, I found only a few minor spelling mistakes that should be corrected. Congratulations to the authors.
Author Response
We want to thank the reviewer for the precious comments and the time spent on the article. The spelling mistakes have been found and corrected.
Reviewer 2 Report
Comments and Suggestions for Authors
The article "Intradevice Repeatability and Interdevice Comparison of Two 2 Specular Microscopy Devices in A Real-Life Setting: Tomey 3 EM-4000 and Nidek CEM-530" from Kecik et al compares the values of endothelial cell density and central corneal thickness obtained by two different specular microscopes.
This is very relevant for ophthalmology clinicians. The paper is well written and the conclusions made are supported by the results.
I have a main concern and question to the authors:
1) Is the data of pleomorphism and polymegathism of both instruments avalable? This comparison will be relevant as well. If the data is available I encourage to add it in Figures 1 and 2.
Minor corrections:
Line 19. "Tomey EM-4000" font size is bigger.
Line 51-54: "The endothelium also plays a role in the metabolism and transport of nutrients like 51 glucose, lactate, and ascorbic acid from the aqueous humor to the cornea, thus regulating 52 the metabolic activity of the corneal epithelium and stroma, which are dependent on the 53 corneal endothelium for their oxygen and nutrient supply (2)". Corneal epithelium can get oxygen as well from the tear film.
Figure 1 resolution is low. Can this be increased? Please explain abbreviations in Figure 1 legend.
Figure 2 resolution is low. Can this be increased?
Line 264. There is an extra period after "values".
Line 287. After "counts" a period is missing and the sentence is unclear.
Author Response
We want to thank the reviewer for the precious comments and the time spent on the article. We would like to address the reviewer's questions and concerns in a point-by-point manner.
- The data concerning the pleomorphism and polmyegatism has unfortunately not been recorded in this study
- The font in the line 19 has been corrected
- The sentence has been modified to read: ‘’… the metabolic activity of the corneal epithelium and stroma, which are dependent, either partially like the former or totally like the latter, on the corneal endothelium for their oxygen and nutrient supply’’
- The abbreviations have been explained as asked. A new better quality figure has been uploaded
- A better quality figure has been uploaded.
- The punctuation has been corrected
- The sentence has been modified to read: ‘’Indeed, a study on corneal donor tissues proposed 5% as a threshold of clinical significance for endothelial cell counts’’
Round 2
Reviewer 2 Report
Comments and Suggestions for Authors
All modifications have been properly addressed.
Author Response
We want to thank the Reviewer for the acknowledgement